# Vibronic effects on the quantum tunnelling of magnetisation in Kramers single-molecule magnets

Andrea Mattioni [1] ✉, Jakob K. Staab[1], William J. A. Blackmore [1], Daniel Reta [1,2,3,4], Jake Iles-Smith[5], Ahsan Nazir [5] & Nicholas F. Chilton [1] ✉

Single-molecule magnets are among the most promising platforms for achieving molecular-scale data storage and processing. Their magnetisation dynamics are determined by the interplay between electronic and vibrational degrees of freedom, which can couple coherently, leading to complex vibronic dynamics. Building on an ab initio description of the electronic and vibrational Hamiltonians, we formulate a non-perturbative vibronic model of the low-energy magnetic degrees of freedom in monometallic single-molecule magnets. Describing their low-temperature magnetism in terms of magnetic polarons, we are able to quantify the vibronic contribution to the quantum tunnelling of the magnetisation, a process that is commonly assumed to be independent of spin-phonon coupling. We find that the formation of magnetic polarons lowers the tunnelling probability in both amorphous and crystalline systems by stabilising the low-lying spin states. This work, thus, shows that spin-phonon coupling subtly influences magnetic relaxation in single-molecule magnets even at extremely low temperatures where no vibrational excitations are present.

Single-molecule magnets (SMMs) hold the potential for realising high-density data storage and quantum information processing[1–4]. These molecules exhibit a ground state comprising two states characterised by a large magnetic moment with opposite orientations, which represents an ideal platform for storing digital data. Slow reorientation of this magnetic moment results in magnetic hysteresis at the single-molecule level at sufficiently low temperatures[5]. The main obstacle to extending this behaviour to room temperature is the coupling of the magnetic degrees of freedom to molecular and lattice vibrations, often referred to as spin–phonon coupling[6]. Thermal excitation of the molecular vibrations causes transitions between different magnetic states, ultimately leading to a complete loss of magnetisation. Advances in the design, synthesis and characterisation of SMMs have shed light on the microscopic mechanisms underlying their desirable

magnetic properties, and have allowed extending the nanomagnet behaviour to increasingly higher temperatures[7–9].

The mechanism responsible for magnetic relaxation in SMMs strongly depends on temperature. At higher temperatures, relaxation is driven by one (Orbach) and two (Raman) phonon transitions between magnetic sublevels[10]. When temperatures approach absolute zero, all vibrations are predominantly found in their ground state. Thus, both Orbach and Raman transitions become negligible and the dominant mechanism is quantum tunnelling of the magnetisation (QTM)[11,12]. This mechanism originates from a coherent coupling between the two magnetic ground states, which leads to the opening of a tunnelling gap. The tunnel coupling allows the population to redistribute between states of opposite magnetisation and thus facilitates magnetic reorientation.

[1]Department of Chemistry, School of Natural Sciences, The University of Manchester, Oxford Road, Manchester M13 9PL, UK. [2]Faculty of Chemistry, The University of the Basque Country UPV/EHU, Donostia 20018, Spain. [3]Donostia International Physics Center (DIPC), Donostia 20018, Spain. [4]IKERBASQUE, Basque Foundation for Science, Bilbao 48013, Spain. [5]Department of Physics and Astronomy, School of Natural Sciences, The University of Manchester, Oxford Road, Manchester M13 9PL, UK. ✉e-mail: andrea.mattioni@manchester.ac.uk; nicholas.chilton@manchester.ac.uk

While the role of vibrations in high-temperature magnetic relaxation is well understood in terms of weak coupling rate equations for the electronic populations[13–16], the connection between QTM and spin–phonon coupling is still largely unexplored. Some analyses have looked at the influence of vibrations on QTM in integer-spin SMMs, where a model spin system was used to show that spin–phonon coupling could open a tunnelling gap[17,18]. However, QTM remains more elusive to grasp in half-integer spin systems, such as monometallic Dy(III) SMMs. In this case, a magnetic field is needed to break the time-reversal symmetry of the molecular Hamiltonian and lift the degeneracy of the ground doublet, as a consequence of Kramers theorem[19]. This magnetic field can be provided by hyperfine interaction with nuclear spins or by dipolar coupling to other SMMs; both these effects have been shown to affect tunnelling behaviour[20–27]. Once the tunnelling gap is opened by a magnetic field, molecular vibrations can in principle affect its magnitude in a nontrivial way (Fig. 1a). In a recent work, Ortu et al. analysed the magnetic hysteresis of a series of Dy(III) SMMs, suggesting that QTM efficiency correlates with molecular flexibility[23]. In another work, hyperfine coupling was proposed to assist QTM by facilitating the interaction between molecular vibrations and spin sublevels[28]. However, a clear and unambiguous demonstration of the influence of spin–phonon coupling on QTM beyond toy-model approaches is still lacking to this date. A reason for this shortfall is found in the common wisdom that vibrations only cause transitions between electronic states when thermally excited, and therefore are unable to influence magnetic relaxation when thermal energy is much lower than their frequency.

In this work, we present a theoretical analysis of the effect of molecular vibrations on the tunnelling dynamics in two prototypical Dy(III) SMMs, $[Dy(Cp^{ttt})_2]^{+}$ [7] and $[Dy(bbpen)Br]$ [29] (Fig. 1b). Our approach is based on a fully ab initio description of the SMM vibrational environment and accounts for the spin–phonon coupling in a non-perturbative way. In this aspect, this work represents a step forward compared to previous theoretical analyses, which relied on a simplified description of phonons as small rotational displacements of the magnetic anisotropy axis and on a standard weak-coupling master equation approach[30]. After deriving an effective low-energy model for the relevant vibronic degrees of freedom based on a Polaron approach[31], we demonstrate that vibrations can either enhance or reduce the quantum tunnelling gap, depending on the orientation of the magnetic field relative to the main anisotropy axis of the SMM. Lastly, we show that different vibrational modes can have competing effects on QTM; depending on how vibrations impact the axiality of the lowest energy magnetic doublet, they can lead to either a decrease or an increase of the tunnelling probability. While identifying vibrations that selectively tune QTM through the chemical design of new SMMs goes beyond the scope of this work, our improved description of vibronic QTM provides a useful framework to articulate further studies in that direction.

## Results

### Ab initio simulations

In this work, we investigate two representative examples of Dy(III) SMMs and explore both amorphous and crystalline phonon environments. The first compound is $[Dy(Cp^{ttt})_2]^{+}$, shown in Fig. 1b, top[7]. It consists of a dysprosium ion Dy(III) enclosed between two negatively charged cyclopentadienyl rings with *tert*-butyl groups at positions 1, 2 and 4 ($Cp^{ttt}$). The crystal field generated by the axial ligands makes the states with larger angular momentum be energetically favourable, resulting in the energy level diagram sketched in Fig. 1a. The energy barrier separating the two degenerate ground states results in magnetic hysteresis, which was observed up to $T = 60$ K[7].

To single out the contribution of molecular vibrations, we focus on a magnetically diluted sample in a frozen solution of dichloromethane (DCM). Thus, our computational model consists of a solvated $[Dy(Cp^{ttt})_2]^{+}$ cation (Fig. 1b, top), which provides a realistic description of the low-frequency vibrational environment, comprised of pseudo-acoustic vibrational modes (Supplementary Note 1). These

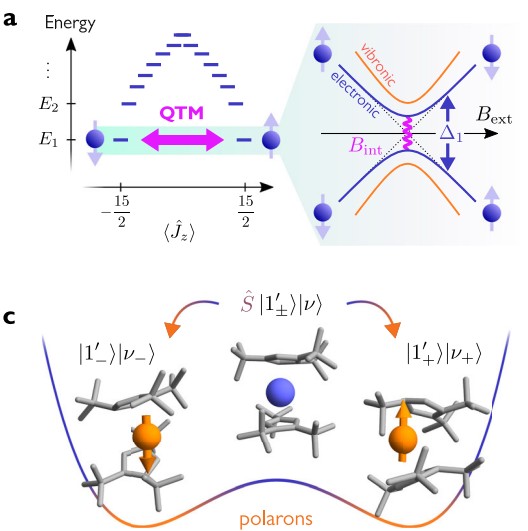

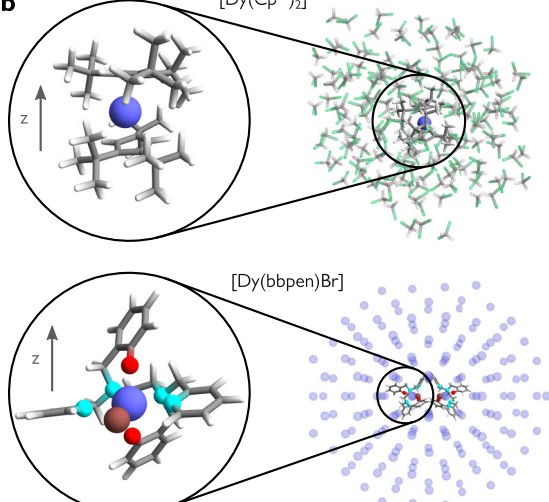

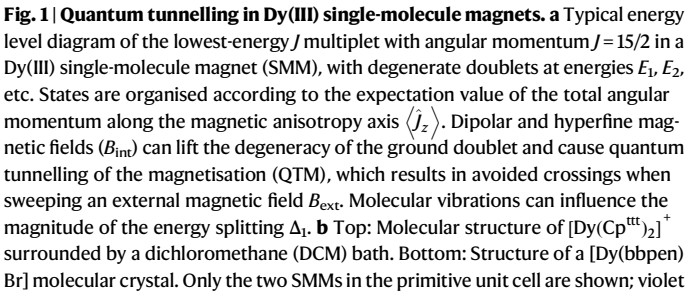

**Fig. 1 | Quantum tunnelling in Dy(III) single-molecule magnets. a** Typical energy level diagram of the lowest-energy $J$ multiplet with angular momentum $J = 15/2$ in a Dy(III) single-molecule magnet (SMM), with degenerate doublets at energies $E_1$, $E_2$, etc. States are organised according to the expectation value of the total angular momentum along the magnetic anisotropy axis $\langle \hat{J}_z \rangle$. Dipolar and hyperfine magnetic fields ($B_{int}$) can lift the degeneracy of the ground doublet and cause quantum tunnelling of the magnetisation (QTM), which results in avoided crossings when sweeping an external magnetic field $B_{ext}$. Molecular vibrations can influence the magnitude of the energy splitting $\Delta_1$. **b** Top: Molecular structure of $[Dy(Cp^{ttt})_2]^{+}$ surrounded by a dichloromethane (DCM) bath. Bottom: Structure of a [Dy(bbpen) Br] molecular crystal. Only the two SMMs in the primitive unit cell are shown; violet spheres represent Dy atoms at other lattice positions. Atoms are colour-coded as follows: Dy (violet), Br (brown), Cl (green), O (red), N (cyan), C (grey), H (white). In both cases, $z$ indicates the direction of the easy axis. **c** Idea behind the polaron transformation $\hat{S}$ of Eq. (6). Each spin state $|1'_\pm\rangle$ is accompanied by a vibrational distortion (greatly exaggerated for visualisation), thus forming a magnetic polaron. Vibrational states $|\nu\rangle$ are now described in terms of harmonic displacements around the deformed structure, which depends on the state of the spin. Polarons provide an accurate physical picture when the spin–phonon coupling is strong and mostly modulates the energy of different spin states but not the coupling between them.

constitute the basis to consider further contributions of dipolar and hyperfine interactions to QTM. Once the equilibrium geometry and vibrational modes of the solvated SMM (which are in general combinations of molecular and solvent vibrations) are obtained at the density-functional level of theory, we proceed to determine the equilibrium electronic structure via complete active space self-consistent field spin–orbit (CASSCF-SO) calculations. The electronic structure is projected onto an effective crystal-field Hamiltonian. The spin–phonon couplings are obtained from a single CASSCF calculation by computing the analytic derivatives of the molecular Hamiltonian with respect to the nuclear coordinates[15]. Further details can be found in the "Methods" section.

The second compound considered in this work is the highly stable [Dy(bbpen)Br] (H$_2$bbpen = *N,N′*-bis(2-hydroxybenzyl)-*N,N′*-bis(2-methylpyridyl)ethylenediamine), shown in Fig. 1b, bottom[29]. It consists of a Dy(III) ion with pentagonal bipyramidal local geometry, with four N and one Br atom coordinating equatorially. Two axially coordinating O atoms give rise to strong easy-axis magnetic anisotropy. The effective barrier for magnetic reversal is around 1000 K and magnetic hysteresis was observed up to 14 K[29]. The small size of the unit cell and the relatively high-symmetry space group ($C222_1$) make this system amenable for spin–phonon coupling calculations in a crystalline environment. The primitive unit cell, consisting of two symmetry-related replicas of [Dy(bbpen)Br], was optimised at the density functional level of theory, and phonons were calculated using a $2 \times 2 \times 1$ supercell expansion. The electronic structure of the Dy(III) centres was obtained with state-average CASSCF-SO and parametrised with a crystal field Hamiltonian. Spin–phonon couplings were obtained via the linear vibronic coupling model[15]. A full account of these methods can be found in ref. 32.

## Polaron model

The lowest-energy angular momentum multiplet of a Dy(III) SMM ($J = 15/2$) can be described by the ab initio vibronic Hamiltonian

$$\hat{H} = \sum_m E_m |m\rangle\langle m| + \sum_j \hat{V}_j \otimes (\hat{b}_j + \hat{b}_j^\dagger) + \sum_j \omega_j \hat{b}_j^\dagger \hat{b}_j, \tag{1}$$

where $E_m$ denotes the energy of the $m$th eigenstate $|m\rangle$ of the crystal field Hamiltonian and $\hat{V}_j \otimes (\hat{b}_j + \hat{b}_j^\dagger)$ represent the spin–phonon coupling operators. The harmonic vibrational modes are described in terms of their bosonic annihilation (creation) operators $\hat{b}_j$ ($\hat{b}_j^\dagger$) and frequencies $\omega_j$.

In the absence of magnetic fields, the Hamiltonian (1) is symmetric under time reversal. This symmetry results in a two-fold degeneracy of the energy levels $E_m$, whose corresponding eigenstates $|m\rangle$ and $|\bar{m}\rangle$ form a time-reversal conjugate Kramers doublet. The degeneracy is lifted by introducing a magnetic field **B**, which couples to the electronic degrees of freedom via the Zeeman interaction $\hat{H}_{\text{Zee}} = \mu_B g_J \mathbf{B} \cdot \hat{\mathbf{J}}$, where $g_J$ is the Landé $g$-factor and $\hat{\mathbf{J}}$ is the total angular momentum operator. To linear order in the magnetic field, each Kramers doublet splits into two energy levels $E_m \pm \Delta_m/2$ corresponding to the states

$$|m_+\rangle = \cos\frac{\theta_m}{2}|m\rangle + e^{i\phi_m}\sin\frac{\theta_m}{2}|\bar{m}\rangle \tag{2}$$

$$|m_-\rangle = -\sin\frac{\theta_m}{2}|m\rangle + e^{i\phi_m}\cos\frac{\theta_m}{2}|\bar{m}\rangle \tag{3}$$

where the energy splitting $\Delta_m$ and the mixing angles $\theta_m$ and $\phi_m$ are determined by the matrix elements of the Zeeman Hamiltonian on the subspace $\{|m\rangle, |\bar{m}\rangle\}$. In addition to the intra-doublet mixing described by Eqs. (2) and (3), the Zeeman interaction also mixes Kramers doublets at different energies. The ground doublet acquires

contributions from higher-lying states

$$|1'_\pm\rangle = |1_\pm\rangle + \sum_{m \neq 1,\bar{1}} |m\rangle \frac{\langle m|\hat{H}_{\text{Zee}}|1_\pm\rangle}{E_1 - E_m} + \mathcal{O}(B^2). \tag{4}$$

These states no longer form a time-reversal conjugate doublet, meaning that the spin–phonon coupling can now contribute to transitions between them.

Since QTM is typically observed at much lower temperatures than the energy gap between the lowest and first excited doublets (which here is ≳600 K[7,29]) we focus on the perturbed ground doublet $|1'_\pm\rangle$. Within this subspace, the Hamiltonian $\hat{H} + \hat{H}_{\text{Zee}}$ takes the form

$$\begin{aligned}
\hat{H}_{\text{eff}} = {} & E_1 + \frac{\Delta_1}{2}\sigma'_z + \sum_j \omega_j \hat{b}_j^\dagger \hat{b}_j \\
& + \sum_j \left(\langle 1|\hat{V}_j|1\rangle - w_j^z \sigma'_z\right)\left(\hat{b}_j + \hat{b}_j^\dagger\right) \\
& - \sum_j \left(w_j^x \sigma'_x + w_j^y \sigma'_y\right)\left(\hat{b}_j + \hat{b}_j^\dagger\right).
\end{aligned} \tag{5}$$

This Hamiltonian describes the interaction between vibrational modes and an effective spin one-half represented by the Pauli matrices $\boldsymbol{\sigma}' = (\sigma'_x, \sigma'_y, \sigma'_z)$, where $\sigma'_z = |1'_+\rangle\langle 1'_+| - |1'_-\rangle\langle 1'_-|$. The vector $\mathbf{w}_j = (\Re\langle 1_-|\hat{W}_j|1_+\rangle, \Im\langle 1_-|\hat{W}_j|1_+\rangle, \langle 1_+|\hat{W}_j|1_+\rangle)$ is defined in terms of the operator $\hat{W}_j = \sum_{m \neq 1,\bar{1}} \hat{V}_j|m\rangle\langle m|\hat{H}_{\text{Zee}}/(E_m - E_1) + \text{h.c.}$, describing the effect of the Zeeman interaction on the spin–phonon coupling. Due to the strong magnetic axiality of the SMM considered here, the longitudinal component of the spin–phonon coupling $w_j^z$ dominates over the transverse part $w_j^x$, $w_j^y$. In this case, we can get a better physical picture of the system by transforming the Hamiltonian (5) to the polaron frame defined by the unitary operator

$$\hat{S} = \exp\left[\sum_{s = \pm} |1'_s\rangle\langle 1'_s| \sum_j \xi_j^s \left(\hat{b}_j^\dagger - \hat{b}_j\right)\right], \tag{6}$$

which mixes electronic and vibrational degrees of freedom by displacing the mode operators by $\xi_j^\pm = (\langle 1|\hat{V}_j|1\rangle \mp w_j^z)/\omega_j$ depending on the state of the effective spin one-half[31]. The idea behind this transformation is to allow nuclei to relax around a new equilibrium geometry, which may be different for every spin state. This lowers the energy of the system and provides a good description of the vibronic eigenstates when the spin–phonon coupling is approximately diagonal in the spin basis (Fig. 1c). In the polaron frame, the longitudinal spin–phonon coupling is fully absorbed into the purely electronic part of the Hamiltonian, while the transverse components can be approximated by their thermal average over vibrations, neglecting their vanishingly small quantum fluctuations (Supplementary Note 2). After transforming back to the original frame, we are left with an effective spin one-half Hamiltonian with no residual spin–phonon coupling $H_{\text{eff}} \approx \hat{H}_{\text{eff}}^{(\text{pol})} + \sum_j \omega_j \hat{b}_j^\dagger \hat{b}_j$, where

$$\hat{H}_{\text{eff}}^{(\text{pol})} = E_1 + \frac{\Delta_1}{2}\sigma''_z + 2\sum_j \frac{\langle 1|\hat{V}_j|1\rangle}{\omega_j} \mathbf{w}_j \cdot \boldsymbol{\sigma}''. \tag{7}$$

The set of Pauli matrices $\boldsymbol{\sigma}'' = \hat{S}^\dagger(\boldsymbol{\sigma}' \otimes \mathbb{1}_{\text{vib}})\hat{S}$ describe the two-level system formed by the magnetic polarons of the form $\hat{S}^\dagger|1'_\pm\rangle|\{\nu_j\}\rangle_{\text{vib}}$, where $\{\nu_j\}$ is a set of occupation numbers for the vibrational modes of the solvent–SMM system. These magnetic polarons can be thought of as magnetic electronic states strongly coupled to a distortion of the molecular geometry. They inherit the magnetic properties of the corresponding electronic states and can be seen as the molecular equivalent of the magnetic polarons observed in a range of magnetic materials[33–35]. Polaron representations of vibronic systems have been employed in a wide variety of settings, ranging from spin-boson

models[31,36] to photosynthetic complexes[37-39], to quantum dots[40-42], providing a convenient basis to describe the dynamics of quantum systems strongly coupled to a vibrational environment. These methods are particularly well suited for condensed matter systems where the electron–phonon coupling is strong but causes very slow transitions between different electronic states, allowing exact treatment of the pure-dephasing part of the electron–phonon coupling and renormalising the electronic parameters. For this reason, the polaron transformation is especially effective for describing our system (Supplementary Note 3). The most striking advantage of this approach is that the average effect of the spin–phonon coupling is included non-perturbatively into the electronic part of the Hamiltonian, leaving behind a vanishingly small residual spin–phonon coupling. As a last step, we bring the Hamiltonian in Eq. (7) into a more familiar form by expressing it in terms of an effective $g$-matrix. We recall that the quantities $\Delta_1$ and $\mathbf{w}_j$ depend linearly on the magnetic field $\mathbf{B}$ via the Zeeman Hamiltonian $\hat{H}_{\mathrm{Zee}}$. An additional dependence on the orientation of the magnetic field comes from the mixing angles $\theta_1$ and $\phi_1$ introduced in Eqs. (2) and (3), appearing in the states $|1_\pm\rangle$ used in the definition of $\mathbf{w}_j$. This further dependence is removed by transforming the Pauli operators back to the basis $\{|1\rangle, |\bar{1}\rangle\}$ via a three-dimensional rotation $\boldsymbol{\sigma} = \mathbf{R}_{\theta_1,\phi_1} \cdot \boldsymbol{\sigma}''$. Finally, we obtain

$$\hat{H}_{\mathrm{eff}}^{(\mathrm{pol})} = E_1 + \mu_B \mathbf{B} \cdot \left( \mathbf{g}^{\mathrm{el}} + \sum_j \mathbf{g}_j^{\mathrm{vib}} \right) \cdot \frac{\boldsymbol{\sigma}}{2}, \qquad (8)$$

for appropriately defined electronic and single-mode vibronic $g$-matrices $\mathbf{g}^{\mathrm{el}}$ and $\mathbf{g}_j^{\mathrm{vib}}$. These are directly related to the electronic splitting term $\Delta_1$ and to the vibronic corrections described by $\mathbf{w}_j$ in Eq. (7), respectively (see Supplementary Note 2 for a thorough derivation). The main advantage of representing the ground Kramers doublet with an effective spin one-half Hamiltonian is that it provides a conceptually simple foundation for studying low-temperature magnetic behaviour of the SMM, confining all microscopic details, including vibronic effects, to an effective $g$-matrix.

## Vibronic modulation of the ground Kramers doublet

We begin by considering the influence of vibrations on the Zeeman splitting of the lowest Kramers doublet. The Zeeman splitting in the absence of vibrations is simply given by $\Delta_1 = \mu_B |\mathbf{B} \cdot \mathbf{g}^{\mathrm{el}}|$. In the presence of vibrations, the electronic $g$-matrix $\mathbf{g}^{\mathrm{el}}$ is modified by adding the vibronic correction $\sum_j \mathbf{g}_j^{\mathrm{vib}}$, resulting in the Zeeman splitting $\Delta_1^{\mathrm{vib}}$. In Fig. 2a we show the Zeeman splittings as a function of the orientation of the magnetic field $\mathbf{B}$ for $[\mathrm{Dy}(\mathrm{Cp}^{\mathrm{ttt}})_2]^+$, parametrised in terms of the polar angles $(\theta, \phi)$. Depending on the field orientation, vibrations can lead to either an increase or decrease of the Zeeman splitting. These changes seem rather small when compared to the largest electronic splitting, obtained when $\mathbf{B}$ is oriented along the $z$-axis (Fig. 1b), as expected for a system with easy-axis anisotropy. However, they become quite significant for field orientations close to the $xy$-plane, where the purely electronic splitting $\Delta_1$ becomes vanishingly small and $\Delta_1^{\mathrm{vib}}$ can be dominated by the vibronic contribution. This is clearly shown in Fig. 2b, c where we decompose the total field $\mathbf{B} = \mathbf{B}_{\mathrm{int}} + \mathbf{B}_{\mathrm{ext}}$ in a fixed internal component $\mathbf{B}_{\mathrm{int}}$ originating from dipolar and hyperfine interactions, responsible for opening a tunnelling gap, and an external part $\mathbf{B}_{\mathrm{ext}}$ which we sweep along a fixed direction across zero. When these fields lie in the plane perpendicular to the purely electronic easy axis, i.e. the hard plane, the vibronic splitting can be three orders of magnitude larger than the electronic one (Fig. 2b). The situation is reversed when the fields lie in the hard plane of the vibronic $g$-matrix (Fig. 2c). We note that this effect is specific to states with easy-axis magnetic anisotropy, however, this is the defining feature of SMMs, such that our results should be generally applicable to all Kramers SMMs. In fact, we observe very similar results for $[\mathrm{Dy}(\mathrm{bbpen})\mathrm{Br}]$ (Supplementary Note 4).

## Internal fields and QTM probability

So far we have seen that spin-phonon coupling can either enhance or reduce the tunnelling gap in the presence of a magnetic field depending on its orientation. For this reason, it is not immediately clear whether its effects survive ensemble averaging in a collection of randomly oriented SMMs, such as for frozen solutions or polycrystalline samples considered in magnetometry experiments. In order to check this, let us consider an ideal field-dependent magnetisation measurement. When sweeping a magnetic field $B_{\mathrm{ext}}$ at a constant rate from positive to negative values along a given direction, QTM is typically observed as a sharp step in the magnetisation of the sample when crossing the region around $B_{\mathrm{ext}} = 0$[11,27]. This sudden change of the magnetisation is due to a non-adiabatic spin-flip transition between the two lowest energy spin states, that occurs when traversing an avoided crossing (see diagram in Fig. 1a, right). The spin-flip probability is given by the celebrated Landau–Zener expression[43-48], which in our case takes the form

$$P_{\mathrm{LZ}} = 1 - \exp\left( -\frac{\pi |\boldsymbol{\Delta}_\perp|^2}{2 |\mathbf{v}|} \right), \qquad (9)$$

where we have defined $\mathbf{v} = \mu_B d\mathbf{B}_{\mathrm{ext}}/dt \cdot \mathbf{g}$, and $\boldsymbol{\Delta}_\perp$ is the component of $\boldsymbol{\Delta} = \mu_B \mathbf{B}_{\mathrm{int}} \cdot \mathbf{g}$ perpendicular to $\mathbf{v}$, while $\mathbf{g}$ denotes the total electronic-vibrational $g$-matrix appearing in Eq. (8) (see Supplementary Note 2 for a derivation of Eq. (9)).

In order to fully characterise the spin–flip process, we need to quantify the internal fields that cause QTM in Kramers SMMs, which originate from either dipolar or hyperfine interactions. In the following, we focus on dipolar fields, since their effects can be observed at much higher temperatures than those required to witness hyperfine interactions (Supplementary Note 5). Samples studied in magnetometry experiments typically contain a macroscopic number of SMMs, each of which produces a microscopic dipole field. We estimate the combined effect of these microscopic dipoles in a $[\mathrm{Dy}(\mathrm{Cp}^{\mathrm{ttt}})_2]^+$ DCM-frozen solution by generating random spatial configurations of SMMs and calculating the resulting field at a specific point in space corresponding to a randomly selected SMM. We repeat this process 10,000 times to obtain the internal field distribution $\mathbf{B}_{\mathrm{int}}$, as shown in Fig. 3a. The orientation of this field is random and its magnitude averages 5.5 mT for a SMM concentration of 170 mM[7] (Supplementary Note 5).

In the case of the $[\mathrm{Dy}(\mathrm{bbpen})\mathrm{Br}]$ molecular crystal, the effect of all Dy atoms within a 100 Å radius of a central magnetic centre was considered in a 5% Dy in Y diamagnetically diluted crystallite[29]. Random Dy/Y substitutions at different sites and random orientations of the magnetising field $\mathbf{B}_{\mathrm{ext}}$ were considered to mimic a powder sample, leading to the distribution shown in Fig. 3b with an average magnitude of 4.9 mT.

We then sample the distribution of internal fields to calculate the corresponding spin-flip probabilities for a randomly oriented SMM using Eq. (9). The effect of spin–phonon coupling on the spin–flip dynamics of an ensemble of SMMs is shown in Fig. 3c, d. The vibronic correction to the ground doublet $g$-matrix leads to a suppression of spin–flip events (orange) compared to a purely electronic model (blue). Despite the significant overlap between the two distributions, spin–phonon coupling results in a ~30% drop in average spin–flip probabilities, represented by the dashed lines in Fig. 3c, d. The vibronic suppression of QTM can be intuitively understood in terms of the polaron energy landscape sketched in Fig. 1c: strong coupling between spin degrees of freedom and molecular distortions can stabilise spin states, introducing a vibrational energy cost for spin reversal; i.e. flipping a spin requires reorganisation of the molecular structure.

From Fig. 3c, d, we also note that crystalline $[\mathrm{Dy}(\mathrm{bbpen})\mathrm{Br}]$ exhibits much larger QTM than $[\mathrm{Dy}(\mathrm{Cp}^{\mathrm{ttt}})_2]^+$ in frozen solution. This can be understood in terms of the different microscopic dipole fields in the two systems. In Supplementary Note 5 we show that $\mathbf{B}_{\mathrm{int}}$ is perfectly

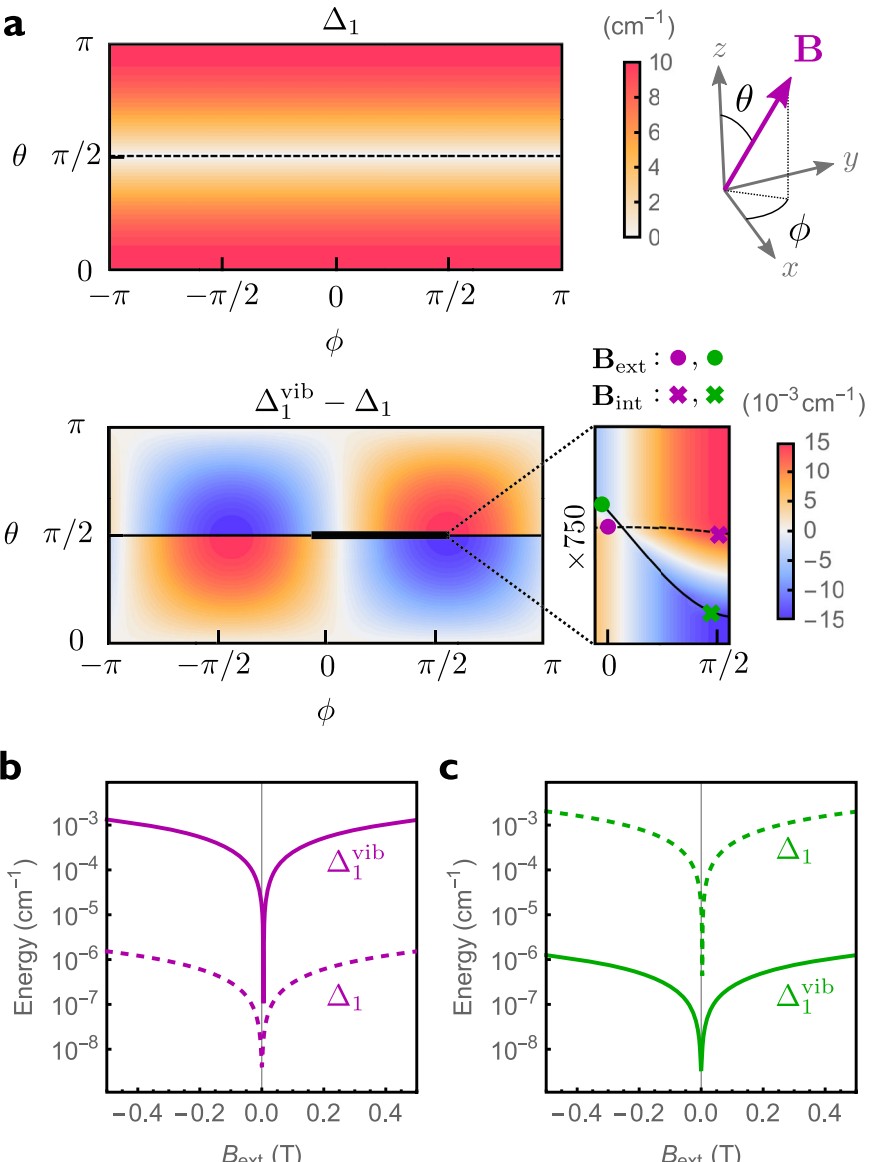

**Fig. 2 | Zeeman splitting of the ground Kramers doublet in [Dy(Cp$^{\text{ttt}}$)$_2$]$^+$.**
**a** Electronic ground doublet splitting ($\Delta_1$, top) and vibronic correction ($\Delta_1^{\text{vib}} - \Delta_1$, bottom) as a function of the orientation of the magnetic field, parametrised in terms of polar and azimuthal angles $\theta$ and $\phi$. The polar angle $\theta$ is measured with respect to the axis joining the cyclopentadienyl centroids, corresponding approximately to the easy axis. The dashed (solid) line corresponds to the electronic (vibronic) hard plane. The magnitude of the magnetic field is fixed to 1 T. **b, c** Electronic (dashed) and vibronic (solid) Zeeman splitting of the ground doublet as a function of the external field magnitude $B_{\text{ext}}$ in the presence of a transverse internal field $B_{\text{int}} = 1$ mT calculated from Eq. (8). External and internal fields are perpendicular to each other and were both chosen to lie in the hard plane of either the electronic (**b**, purple) or vibronic (**c**, green) $g$-matrix. The orientation of the external (internal) field is shown for both cases as circles (crosses) in the inset in (**a**), with colours matching the ones in (**b**) and (**c**).

isotropic in a frozen solution. On the contrary, due to the symmetry of the [Dy(bbpen)Br] molecular crystal, the component of the internal field along the intra-unit cell Dy–Dy direction survives orientational averaging, resulting in an average transverse component of 1.2 mT (Supplementary Note 5).

## Discussion

As shown above, the combined effect of all vibrations in a randomly oriented ensemble of SMMs is to reduce QTM. However, not all vibrations contribute to the same extent. Based on the polaron model introduced above, vibrations with large spin–phonon coupling and low frequency have a larger impact on the magnetic properties of the ground Kramers doublet. This can be seen from Eq. (7), where the vibronic correction to the effective ground Kramers Hamiltonian is weighted by the factor $\langle 1|\hat{V}_j|1\rangle/\omega_j$. Another property

of vibrations that can influence QTM is their symmetry. In monometallic SMMs, QTM has generally been correlated with a reduction of axial symmetry, either by the presence of flexible ligands or by transverse magnetic fields. Since we are interested in symmetry only as long as it influences magnetism, it is useful to introduce a measure of axiality on the $g$-matrix, such as

$$A(\mathbf{g}) = \frac{\parallel \mathbf{g} - \frac{1}{3}\text{Tr}\mathbf{g} \parallel}{\sqrt{\frac{2}{3}}\text{Tr}\mathbf{g}}, \tag{10}$$

where $\parallel \cdot \parallel$ denotes the Frobenius norm. This measure yields 1 for perfect easy-axis anisotropy, 1/2 for an easy-plane system, and 0 for the perfectly isotropic case. The axiality of an individual vibrational mode can be quantified as $A_j = A(\mathbf{g}^{\text{el}} + \mathbf{g}_j^{\text{vib}})$ by building a single-mode vibronic

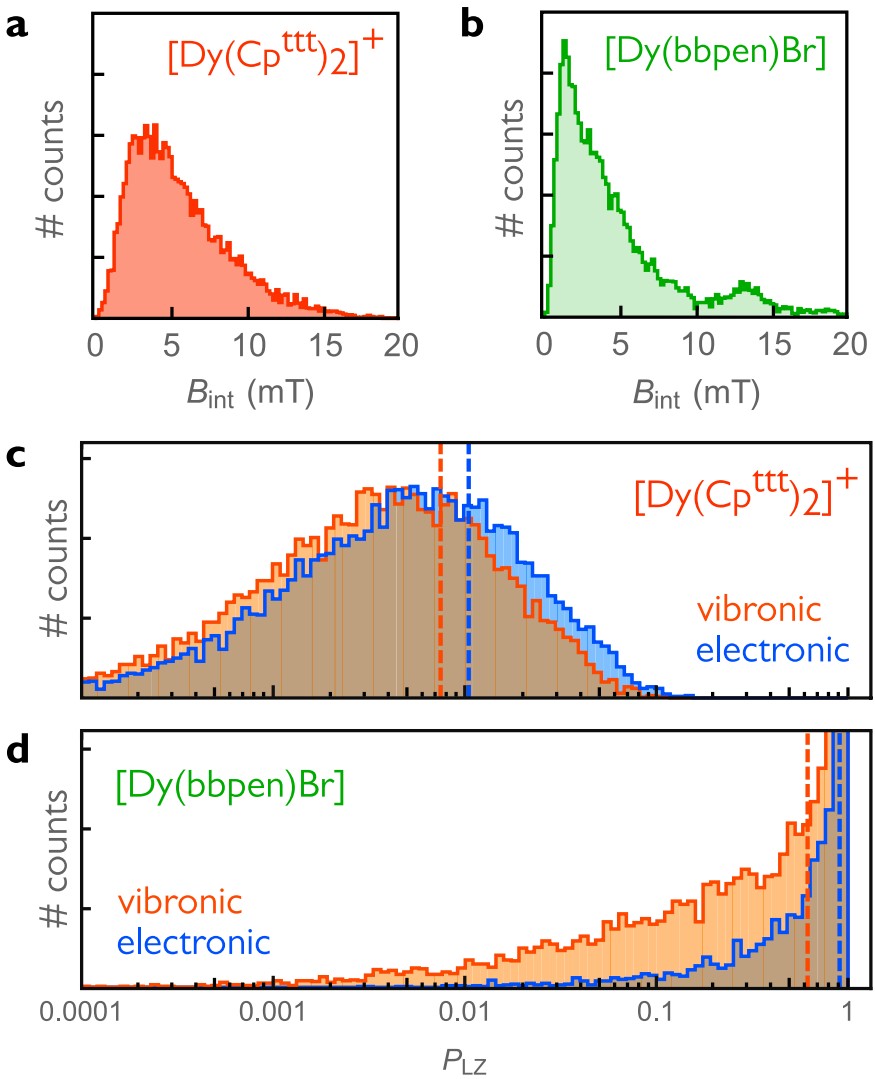

**Fig. 3 | Internal fields and spin-flip probability. a, b** Distribution of internal field magnitudes $B_{int}$ experienced by a Dy centre due to the dipolar fields produced by surrounding Dy centres, magnetised by a randomly oriented external field $\mathbf{B}_{ext}$. For $[Dy(Cp^{ttt})_2]^+$ (**a**), a uniform spatial distribution of 1000 randomly oriented single-molecule magnets (SMMs) around a central Dy(III) was assumed, corresponding to a 170 mM solution in dichloromethane. For the [Dy(bbpen)Br] molecular crystal (**b**), we considered the total dipolar field arising from all Dy centres within a 100 Å radius from a central Dy assuming 5% diamagnetic dilution. **c, d** Distribution of electronic (blue) and vibronic (orange) Landau–Zener spin–flip probabilities $P_{LZ}$, calculated for a randomly oriented SMM subjected to the dipolar fields shown above, assuming an external field sweep rate of 10 Oe/s. Average values are shown as dashed lines: (**c**) 0.0104 (blue) and 0.0074 (orange); (**d**) 0.903 (blue) and 0.618 (orange). All histograms are obtained from an ensemble of 10,000 random external field orientations and dipole arrangements.

$g$-matrix, analogous to the multi-mode one introduced in Eq. (8). We might be tempted to intuitively conclude that polaron formation always increases the axiality with respect to its electronic value $A_{el} = A(\mathbf{g}^{el})$, given that the collective effect of the spin–phonon coupling is to reduce QTM. However, when considered individually, some vibrations can have the opposite effect of effectively reducing the magnetic axiality.

In order to see how axiality correlates to QTM, we calculate the single-mode spin-flip probabilities $\langle P_j \rangle$. These are obtained by replacing the multi-mode vibronic $g$-matrix in Eq. (8) with the single-mode one $\mathbf{g}^{el} + \mathbf{g}_j^{vib}$, and following the same procedure detailed in Supplementary Note 2. The single-mode contribution to the spin–flip probability unambiguously correlates with mode axiality, as shown in Fig. 4a for $[Dy(Cp^{ttt})_2]^+$; the correlation is even starker for crystalline [Dy(bbpen)Br] (Fig. 4c). Vibrational modes that lead to a larger QTM probability are likely to reduce the magnetic axiality (top-left sector). Vice versa, those vibrational modes that enhance axiality also suppress QTM (bottom-right sector).

As a first step towards uncovering the microscopic basis of this unexpected behaviour, we single out the vibrational modes that have the largest impact on magnetic axiality in both directions. These vibrational modes, labelled A, B for $[Dy(Cp^{ttt})_2]^+$ and C, D for [Dy(bbpen)Br], represent a range of qualitatively distinct vibrations, as can be observed in Fig. 4b, d. In the case of $[Dy(Cp^{ttt})_2]^+$, mode A is mainly localised on one of the $Cp^{ttt}$ ligands and features atomic displacements predominantly perpendicular to the easy axis. Mode B, on the other hand, involves axial distortions of the Cp rings and, to a lesser extent, rotations of the methyl groups. Thus, it makes sense intuitively that A would lead to an increased QTM probability, while the opposite is true for B, as observed in Fig. 4a.

However, the connection between the magnetic axiality defined in Eq. (10) and vibrational motion is not always straightforward. In the case of [Dy(bbpen)Br], mode C mainly involves a tilt of the two equatorial pyridyl groups. This movement disrupts axiality and enhances QTM. On the other hand, mode D features equatorial motion of the first coordination sphere of the Dy(III) ion, involving the movement of

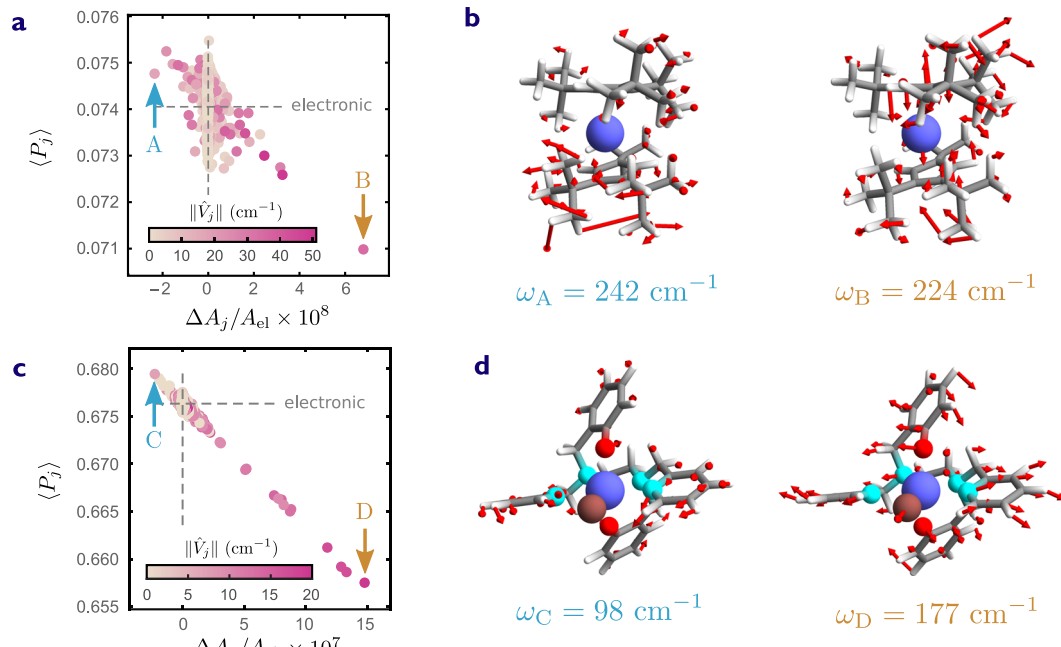

**Fig. 4 | Single-mode contributions to tunnelling of the magnetisation.**
**a**, **c** Single-mode vibronic spin-flip probabilities plotted for each vibrational mode, shown as a function of the mode axiality $\Delta A_j = A_j - A_{el}$ relative to the electronic axiality $A_{el}$. The magnitude of the internal field is fixed to $B_{int} = 1\,mT$ and the external field sweep rate is 10 Oe/s. The probabilities $\langle P_j \rangle$ are obtained by averaging over random orientations of external and internal fields. The colour coding represents the spin–phonon coupling strength $\| \hat{V}_j \|$. Grey dashed lines corresponds to a purely electronic model. **a** and **c** correspond to amorphous $[Dy(Cp^{ttt})_2]^+$ and crystalline $[Dy(bbpen)Br]$. **b**, **d** Visual representation of the displacements induced by the vibrational modes indicated by arrows in **a** and **c** denoted by A–D; the corresponding vibrational frequencies are denoted by $\omega_A, \omega_B, \omega_C, \omega_D$.

Br and Dy itself in the hard plane. However, this vibrational mode induces a suppression of QTM, as seen in Fig. 4c, rather than an increase, as would be expected based on the above symmetry arguments. This shows that $\Delta A_j$ does not necessarily correlate to atomic motions, but can be a useful proxy for determining a given vibration's contribution to the QTM probability. In fact, the correlation between the two quantities can be rationalised with the help of the simple toy model presented in Supplementary Note 6. Nonetheless, we note that the out-of-phase motion of the equatorial pyridyl groups in D preserves axiality and could contribute to its efficiency at suppressing QTM. It is also worth noting that Briganti et al. recently demonstrated that the motion of atoms beyond the first coordination sphere of the central Dy(III) ion can greatly influence spin dynamics in the Raman regime through bond polarisation effects[14]. Performing a similar electrostatic analysis in the context of our polaron model is beyond the scope of this work; however, it represents an interesting direction for further investigations elucidating the role of vibrations on QTM.

In conclusion, we have presented a detailed description of the effect of molecular and solvent vibrations on the quantum tunnelling between low-energy spin states in two different single-ion Dy(III) SMMs, corresponding to amorphous and crystalline environments. Our theoretical results, based on an ab initio approach, are complemented by a polaron treatment of the relevant vibronic degrees of freedom, which does not suffer from any weak spin-phonon coupling assumption and is therefore well-suited to other strong coupling scenarios. We have been able to derive a non-perturbative vibronic correction to the effective g-matrix of the lowest-energy Kramers doublet, which we have used as a basis to determine the tunnelling dynamics in an idealised magnetic field sweep experiment, building on Landau–Zener theory. This has allowed us to formulate the observation that spin-phonon coupling does have an influence on QTM, albeit a subtle one (~30%), as opposed to the widespread belief that magnetic tunnelling is not influenced by vibrations since it only becomes effective at low temperatures. This effect is rooted in the formation of

magnetic polarons, which results in a redefinition of the magnetic anisotropy of the ground Kramers doublet. Our theoretical treatment is fully ab initio and represents a significant improvement over other theoretical descriptions of QTM which rely on weak coupling assumptions. Lastly, we observe that specific vibrational modes can either enhance or suppress QTM. This behaviour correlates to the magnetic axiality of each mode, which can be used as a proxy for determining whether a specific vibration enhances or hinders tunnelling. Our analysis suggests that there may be a positive side to spin–phonon coupling in QTM. Enhancing the coupling to specific vibrations via appropriate chemical design while keeping detrimental vibrations under control, could in principle increase magnetic axiality and thus suppress QTM even further. However, translating this observation into clear-cut chemical design guidelines remains an open question, that requires the analysis of other molecular systems. As ab initio spin–phonon coupling calculations become more accessible, the approach presented here can be applied to the study of vibronic QTM in other SMMs and thus represents a valuable tool for understanding the role of vibrations in low-temperature magnetic relaxation.

## Methods
The ab initio model of the DCM-solvated $[Dy(Cp^{ttt})_2]^+$ molecule is constructed using a multi-layer approach. During geometry optimisation and frequency calculation, the system is partitioned into two layers following the ONIOM scheme[49]. The high-level layer, consisting of the SMM itself and the first solvation shell of 26 DCM molecules, is described by density functional theory (DFT) while the outer bulk of the DCM ball constitutes the low-level layer modelled by the semi-empirical PM6 method. All DFT calculations are carried out using the pure PBE exchange-correlation functional[50] with Grimme's D3 dispersion correction. Dysprosium is replaced by its diamagnetic analogue yttrium for which the Stuttgart RSC 1997 ECP basis is employed[51]. Cp ring carbons directly coordinated to the central ion are equipped with Dunning's correlation consistent triple-zeta polarised cc-pVTZ basis

set and all remaining atoms with its double-zeta analogue cc-pVDZ[52]. Subsequently, the electronic spin states and spin-phonon coupling parameters are calculated at the CASSCF-SO level explicitly accounting for the strong static correlation present in the f-shell of Dy(III) ions. At this level, environmental effects are treated using an electrostatic point charge representation of all DCM atoms. All DFT/PM6 calculations are carried out with GAUSSIAN version 9 revision D.01[53] and the CASSCF calculations are carried out with OPENMOLCAS version 21.06[54].

The starting $[Dy(Cp^{ttt})_2]^+$ solvated system was obtained using the solvate programme belonging to the AmberTool suite of packages, with box as the method and CHCL3BOX as the solvent model. Chloroform molecules were subsequently converted to DCM. From this large system, only molecules falling within 9 Å from the central metal atom are considered from now on. The initial disordered system of 160 DCM molecules packed around the $[Dy(Cp^{ttt})_2]^+$ crystal structure[7] is pre-optimised in steps, starting by only optimising the high-level layer atoms and freezing the rest of the system. The low-layer atoms are pre-optimised along the same lines starting with DCM molecules closest to the SMM and working in shells towards the outside. Subsequently, the whole system is geometry optimised until RMS (maximum) values in force and displacement corresponding to 0.00045 au (0.0003 au) and 0.0018 au (0.0012 au) are reached, respectively. After adjusting the isotopic mass of yttrium to that of dysprosium $m_{Dy} = 162.5$ u, vibrational normal modes and frequencies of the entire molecular aggregate are computed within the harmonic approximation.

Electrostatic atomic point charge representations of the environment DCM molecules are evaluated for each isolated solvent molecule independently at the DFT level of theory employing the CHarges from ELectrostatic Potentials using a Grid-based (ChelpG) method[55], which serve as a classical model of environmental effects in the subsequent CASSCF calculations.

The evaluation of equilibrium electronic states and spin-phonon coupling parameters is carried out at the CASSCF level including scalar relativistic effects using the second-order Douglas–Kroll Hamiltonian and spin–orbit coupling through the atomic mean field approximation implemented in the restricted active space state interaction approach[56,57]. The dysprosium atom is equipped with the ANO-RCC-VTZP, the Cp ring carbons with the ANO-RCC-VDZP and the remaining atoms with the ANO-RCC-VDZ basis set[58]. The resolution of the identity approximation with an on-the-fly acCD auxiliary basis is employed to handle the two-electron integrals[59]. The active space of 9 electrons in 7 orbitals, spanned by $4f$ atomic orbitals, is employed in a state-average CASSCF calculation including the 18 lowest lying sextet roots which span the ${}^6H$ and ${}^6F$ atomic terms.

We use our own implementation of spin Hamiltonian parameter projection to obtain the crystal field parameters $B_k^q$ entering the Hamiltonian

$$\hat{H}_{CF} = \sum_{k=2,4,6} \sum_{q=-k}^{k} \theta_k B_k^q O_k^q(\hat{\mathbf{J}}),\qquad(11)$$

describing the ${}^6H_{15/2}$ ground state multiplet. Operator equivalent factors and Stevens operators are denoted by $\theta_k$ and $O_k^q(\hat{\mathbf{J}})$, where $\hat{\mathbf{J}} = (\hat{J}_x \hat{J}_y \hat{J}_z)$ are the angular momentum components. Spin–phonon coupling arises from changes to the Hamiltonian (11) due to slight distortions of the molecular geometry, parametrised as

$$B_k^q(\{X_j\}) = B_k^q + \sum_{j=1}^{M} \frac{\partial B_k^q}{\partial X_j} X_j + \dots,\qquad(12)$$

where $X_j$ denotes the dimensionless $j$th normal coordinate of the molecular aggregate. The derivatives $\partial B_k^q / \partial X_j$ are calculated using the linear vibronic coupling (LVC) approach described in ref. 15 based on the state-average CASSCF density-fitting gradients and non-adiabatic coupling involving all 18 sextet roots. Finally, we express the dimensionless normal coordinates in terms of bosonic creation and annihilation operators as $\hat{X}_j = (\hat{b}_j + \hat{b}_j^\dagger)/\sqrt{2}$, which define the system part of the spin–phonon coupling operators in Eq. (1) as

$$\hat{V}_j = \frac{1}{\sqrt{2}} \sum_{k,q} \theta_k \frac{\partial B_k^q}{\partial X_j} O_k^q(\hat{\mathbf{J}}).\qquad(13)$$

## Reporting summary
Further information on research design is available in the Nature Portfolio Reporting Summary linked to this article.

## Data availability
The data generated in this study have been deposited in the Figshare database and can be accessed at https://doi.org/10.48420/21892887[60]. Source data for all figures are provided with this paper. Source data are provided with this paper.

## Code availability
The code used to calculate ab initio spin-phonon couplings is part of our in-house Python packages `spin_phonon_suite` and `angmom_suite`, freely available from the PyPI repository at https://pypi.org/project/spin-phonon-suite/ and https://pypi.org/project/angmom-suite/.

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

## Acknowledgements

This work was made possible thanks to the ERC grant 2019-STG-851504 and Royal Society fellowship URF191320 (N.F.C.). The authors acknowledge support from the Computational Shared Facility at the University of Manchester.

## Author contributions

A.M. formulated and implemented the effective polaron model with input from J.I.-S. and A.N.; J.K.S. and D.R. performed the ab initio calculations with guidance from N.F.C.; A.M. estimated dipolar fields with input from W.J.A.B.; N.F.C. supervised the work. All authors contributed towards the analysis, discussions and preparation of the manuscript.

## Competing interests

The authors declare no competing interests.
