## [Peer Review File · Nature Communications]

Reviewers' Comments:

Reviewer #1:

Remarks to the Author:

Mattioni et al. present a theoretical model to describe tunneling demagnetization in Kramers' doublet single molecule magnets. The model is based on a "polaronic" description of the coupling between vibrations and the ground doublet, where the tunneling probability is facilitated by the dipolar field stemming from the environment. Authors conclude that some vibrations can suppress tunneling although the overall effect is the promotion of demagnetization. This conclusion is presented as a key result of this study but there is no surprise on this since vibrations can either lower or rise magnetic axiality, as already known from theoretical models to treat Raman/Orbach demagnetization.

In my opinion, the most interesting result is the ability to predict tunneling probabilities straight from ab initio data. To achieve this, authors estimate the dipolar field from a back of the envelope calculation (S5) which contrasts with the highly sophisticated treatment given to the electronic and vibrational wave functions. Furthermore, there is no specific prediction for the tunneling probability but only an external field range consistent with expectations. Authors must make an effort to better estimate this dipolar field.

First, they must move the discussion S5 to the main text, as this is key to the model. Secondly, they must provide a concrete number for the model prediction of P (with and without vibrations). Finally, and more importantly, authors must calculate the dipolar field by more sophisticated methods to see if the potentially accurate predictions produced by this model hold. In this sense, I suggest to explore the work of Giulia Galli and Alessandro Lunghi and other authors in this field.

Reviewer #2:

Remarks to the Author:

The manuscript by Mattioni et al. points towards the issue of up to what extent the relevant vibration bath can promote QTM. While this open problem is appealing as QTM is mostly regarded as just driven by internal magnetic fields, incorrect assumptions and conclusions are done. So, major revisions (see below) are needed that may preclude to publish this contribution in Nature Communications.

Introduction (minor revisions):

In "These molecules exhibit a doubly-degenerate ground state, comprising," remove "doubly-degenerate" (non-Kramers nanomagnets do not necessarily have a degenerate ground doublet) and ",".

Change "magnetic properties, extending this behaviour" to "magnetic properties, and have allowed extending the nanomagnet behaviour"

In "Thus, both Orbach and Raman transitions become negligible and (...) (QTM) between the two degenerate ground states. (...) the two otherwise degenerate ground states," what if the two ground states are non-degenerate? Please, rephrase.

The sentence "This process relies on the presence of a coherent coupling (..) thus leading to facile magnetic reorientation." is too large, rough, and should be completely rephrased.

Please, avoid using "verb+ing" after "," so many times, specially if "verb+ing" is not followed by a predicate. This happens more times in the rest of the manuscript. I will not refer to them anymore but, please, proceed to an comprehensive correction.

The alternate use between the words "SMM" and "complex" is rather confusing Please, unify these terms.

Could the authors check the sentence "However, QTM remains more elusive to grasp in half-integer

spin complexes, (...) since it is observed experimentally despite being forbidden by Kramers theorem [18]." Why QTM is forbidden for half-integer spin systems (which have a degenerate ground doublet at zero field) in virtue of the Kramers theorem?

"and discuss first steps in that direction." -> "and discuss first steps to exploit chemical design in that direction."

---Methods (minor revisions):

"makes the states with larger angular momentum energetically favourable" -> "makes the states with larger angular momentum be energetically favourable"

I guess that the operators " V_j " in Eq. (1) describe the electronic spin degree of freedom. Thus, " V_j represent the spin-phonon coupling operators" should be replaced by " $V_j \otimes (b_j + b_j^\dagger)$ represent the spin-phonon coupling operators".

---Results (major revisions):

What are the two shaded pale-red areas in Figure 3? Please, explain in the caption.

-Regarding "In Fig. 3, we observe that the values of B_{int} required by the vibronic model to reproduce the observed spin-flip probability are perfectly consistent with the dipolar fields naturally occurring in the sample, whereas the purely electronic model necessitates internal fields that are one order of magnitude larger.":

I think this sentence is crucial and should be further elaborated for a proper benchmarking of the model developed by the authors when applied to their case study. According to magnetisation measurements, the authors expect an internal field B_{int} in their case study system around 0.001 T if their hypothesis (namely, the non-vibronic-driven QTM must be complemented by the vibronic-driven QTM to explain the experimental value $\langle PLZ \rangle = 0.27$) is true. On the other hand, in case vibronic-driven QTM were actually non-existent, B_{int} should lie a bit below 0.01 T to still explain the value $\langle PLZ \rangle = 0.27$.

Is it possible to somehow compute B_{int} or to have a direct experimental access to it? If so, how does this computed or directly experimentally-determined B_{int} value compare with 0.001 and 0.01 T?

-Regarding "Despite leading to an overall increase of the spin-flip probability on average, coupling the spin to specific vibrations can increase the magnetic axiality of the complex and suppress QTM. This opens a new avenue for the improvement of magnetic relaxation times in SMMs, shifting the role of vibrations from purely antagonistic to potentially beneficial.":

I am afraid it is not that simple for the general case. One could re-design a given molecule to suppress a specific detrimental mode or even to make a specific beneficial mode appear. But, what the authors seem to miss here is the fact that vibrations are not independent: the modification of the said molecule may alter the whole vibration spectrum and nobody can guarantee that (i) the rest of vibrations remain unaltered and (ii) new detrimental modes do not take place. All in all, the modification of the molecule with the intention of only addressing a single mode could lead to also modify other modes -or even make new ones appear- in such a way that the QTM is even larger than before the said modification.

-Regarding the last paragraph "According to the results shown above, the ideal candidates to observe vibronic suppression of QTM (...)":

it happens to be rather poorly elaborated (and so the Conclusions section) in terms of a clear connection to chemical design. Moreover, the proposed requirements which the vibrations should fulfill seem to be even quite demanding. While low-frequency and strongly-coupled-to-the-spin vibrations may help to suppress QTM, they are precisely the ones that contribute to accelerate the thermally-activated relaxation mechanisms.

Reviewer #3:

Remarks to the Author:

Authors present a detailed theoretical study on the effects of local vibrations onto the magnetic tunneling in a prototypical case of molecular magnets. The problem is quite subtle and it requires the understanding of many complementary effects (magnetic anisotropy, vibrations, coupling spin phonons, effects of magnetic fields on the quantum tunneling etc.) at the same time. This is done in a very extensive way by the authors who consider the case of $[\text{Dy}(\text{Cpttt})_2]^+$ in details including different configurations of the solvent and for dipolar/hyperfine interactions. That is remarkable. Besides that, authors develop a non-perturbative vibronic approach that allows to treat the problem also in the case of strong spin-phonon coupling regime. This also goes beyond the state of the art. The -indirect- effect of vibrations on the quantum tunneling of magnetization is clearly highlighted and parametrized by the introduction of the measure of the axiality. This aspect is not yet treated in such a detailed way and it represents a clear step ahead in the literature. Method is holistic, it looks correct and sufficiently innovative. Presentation is clear and it duly takes into account previous publications. Results are not very surprising and probably not easily exportable to other molecules/ligands. The conclusion that molecular engineering could be driven by these results is in my opinion too optimistic and maybe misleading.

Overall, my opinion is that this work deserves attention and merits publication.

Few suggestions for the authors:

-Since the complexity of the problem will soon diverge with the size of the molecule, it seems doubtful that this approach can be implanted to other cases, therefore it seems more fair to present this work as "case study" on a prototypical Dy(III) SMM rather than a general approach. Authors should consider this in revising the key sessions of the manuscript (abstract, introduction, conclusions)

- Identify vibrations is the tricky point yet the main suggestion of the work. It would be interesting to spell out the low energy spectrum of the Dy(III) case considered here, for instance by zooming fig.1c (for instance 0-50 cm^{-1}). If this zoom looks favorable, it could be interesting to quench "bad" vibrational mode and keep the "good" one by decreasing the temperature and this may stimulate some experiments.

Reply to reviewers' comments

Reviewer 1:

Mattioni et al. present a theoretical model to describe tunneling demagnetization in Kramers' doublet single molecule magnets. The model is based on a "polaronic" description of the coupling between vibrations and the ground doublet, where the tunneling probability is facilitated by the dipolar field stemming from the environment. Authors conclude that some vibrations can suppress tunneling although the overall effect is the promotion of demagnetization. This conclusion is presented as a key result of this study but there is no surprise on this since vibrations can either lower or rise magnetic axiality, as already known from theoretical models to treat Raman/Orbach demagnetization.

We note that our conclusions are now different in the revised manuscript. However, we disagree the referee that our work "is no surprise": there has been, prior to this work, no direct interrogation of how vibrations influence QTM. Typical descriptions of direct transitions between Kramers states found in the literature, e.g. Ho, Chibotaru, Phys. Rev. Lett. B 97, 024427 (2018), rely on several approximations that limit their scope. Most notably, low-frequency vibrations are limited to small rotational displacements of the magnetic anisotropy axis, and the description of their effect on spin dynamics is based on a master equation approach and is thus inherently perturbative. In this work, we contrast such treatment with a non-perturbative model based on a fully *ab initio* description of phonons. In our approach, phonons lead to a significant renormalisation of the magnetic anisotropy, rather than just causing weak transitions between purely electronic states. We highlighted these differences by amending the introduction and the conclusions sections of the manuscript.

In my opinion, the most interesting result is the ability to predict tunneling probabilities straight from *ab initio* data. To achieve this, authors estimate the dipolar field from a back of the envelope calculation (S5) which contrasts with the highly sophisticated treatment given to the electronic and vibrational wave functions. Furthermore, there is no specific prediction for the tunneling probability but only an external field range consistent with expectations. Authors must make an effort to better estimate this dipolar field.

First, they must move the discussion S5 to the main text, as this is key to the model. Secondly, they must provide a concrete number for the model prediction of P (with and without vibrations). Finally, and more importantly, authors must calculate the dipolar field by more sophisticated methods to see if the potentially accurate predictions produced by this model hold. In this sense, I suggest to explore the work of Giulia Galli and Alessandro Lunghi and other authors in this field.

We thank the reviewer for raising this point. We significantly improved our estimate of the internal dipolar fields by performing some Monte Carlo simulations. We have completely redesigned Fig. 3 and reformulated the discussion around it accordingly. We now provide specific values for tunnelling probabilities and internal fields and show their full probability distribution. We have expanded Section S6 with further details on the internal field estimate.

Reviewer 2:

The manuscript by Mattioni et al. points towards the issue of up to what extent the relevant vibration bath can promote QTM. While this open problem is appealing as QTM is mostly regarded as just driven by internal magnetic fields, incorrect assumptions and conclusions are done. So, major revisions (see below) are needed that may preclude to publish this contribution in Nature Communications.

This manuscript has undergone a major revision, and has been considerably expanded, as explained at the beginning of this document. We hope that the reviewer finds this version more tempered.

Introduction (minor revisions):

In "These molecules exhibit a doubly-degenerate ground state, comprising," remove "doubly-degenerate" (non-Kramers nanomagnets do not necessarily have a degenerate ground doublet) and ",".

Change "magnetic properties, extending this behaviour" to "magnetic properties, and have allowed extending the nanomagnet behaviour"

In "Thus, both Orbach and Raman transitions become negligible and (...) (QTM) between the two degenerate ground states. (...) the two otherwise degenerate ground states," what if the two ground states are non-degenerate? Please, rephrase.

The sentence "This process relies on the presence of a coherent coupling (..) thus leading to facile magnetic reorientation." is too large, rough, and should be completely rephrased.

Please, avoid using "verb+ing" after "," so many times, specially if "verb+ing" is not followed by a predicate. This happens more times in the rest of the manuscript. I will not refer to them anymore but, please, proceed to an comprehensive correction.

The alternate use between the words "SMM" and "complex" is rather confusing Please, unify these terms.

We implemented the changes requested above and rephrased some sentences following the reviewer's suggestions. Moreover, substantial parts of the manuscript have been completely rewritten in light of the revised results and conclusions.

Could the authors check the sentence "However, QTM remains more elusive to grasp in half-integer spin complexes, (...) since it is observed experimentally despite being forbidden by Kramers theorem [18]." Why QTM is forbidden for half-integer spin systems (which have a degenerate ground doublet at zero field) in virtue of the Kramers theorem?

Kramers theorem states that a system possessing time-reversal symmetry (i.e. no magnetic fields) with an odd number of electrons, like a Dy(III) ion, exhibits doubly degenerate energy levels. In order to initiate population transfer dynamics in a system initially prepared in one of the two degenerate ground states, we need to break time-reversal symmetry (i.e. apply a magnetic field), which, in turn, lifts the degeneracy (i.e. opens a tunnelling gap). In other words, tunnelling is allowed in Kramers systems only in the presence of a magnetic field. We rephrased this sentence and the following one in the hope of clarifying this point.

"and discuss first steps in that direction." -> "and discuss first steps to exploit chemical design in that direction."

Considering further comments from reviewers 2 and 3, we have toned down the discussion on chemical design, and consequently reformulated this sentence altogether.

---Methods (minor revisions):

"makes the states with larger angular momentum energetically favourable" -> "makes the states with larger angular momentum be energetically favourable"

I guess that the operators " V_j " in Eq. (1) describe the electronic spin degree of freedom. Thus, " V_j represent the spin-phonon coupling operators" should be replaced by " $V_j \otimes (b_j + b_j^\dagger)$ represent the spin-phonon coupling operators".

We have now implemented these two changes.

---Results (major revisions):

What are the two shaded pale-red areas in Figure 3? Please, explain in the caption.

Following the comments of reviewers 1 and 3, we have improved our estimates of dipolar fields and completely redesigned Fig. 3 accordingly (see next comment).

-Regarding "In Fig. 3, we observe that the values of B_{int} required by the vibronic model to reproduce the observed spin-flip probability are perfectly consistent with the dipolar fields naturally occurring in the sample, whereas the purely electronic model necessitates internal fields that are one order of magnitude larger.":

I think this sentence is crucial and should be further elaborated for a proper benchmarking of the model developed by the authors when applied to their case study. According to magnetisation measurements, the authors expect an internal field B_{int} in their case study system around 0.001 T if their hypothesis (namely, the non-vibronic-driven QTM must be complemented by the vibronic-driven QTM to explain the experimental value ≈ 0.27) is true. On the other hand, in case vibronic-driven QTM were actually non-existent, B_{int} should lie a bit below 0.01 T to still explain the value ≈ 0.27 .

Is it possible to somehow compute B_{int} or to have a direct experimental access to it? If so, how does this computed or directly experimentally-determined B_{int} value compare with 0.001 and 0.01 T?

We thank the reviewer for the useful suggestion. Based on this and on the comments of reviewers 1 and 3, we have completely reformulated the discussion on internal fields by employing a more accurate estimate, as we explain in our reply to reviewer 1's second comment. Figure 3 has been updated accordingly. Section S6 offers further details. Moreover, following the major revision of this work's results, we have decided to remove the discussion on experimental determination of QTM probabilities.

-Regarding "Despite leading to an overall increase of the spin-flip probability on average, coupling the spin to specific vibrations can increase the magnetic axiality of the complex and suppress QTM. This opens a new avenue for the improvement of magnetic relaxation times in SMMs, shifting the role of vibrations from purely antagonistic to potentially beneficial.":

I am afraid it is not that simple for the general case. One could re-design a given molecule to suppress a specific detrimental mode or even to make a specific beneficial mode appear. But, what the authors seem to miss here is the fact that vibrations are not independent: the modification of the said molecule may alter the whole vibration spectrum and nobody can guarantee that (i) the rest of vibrations remain unaltered and (ii) new detrimental modes do not take place. All in all, the modification of the molecule with the intention of only addressing a single mode could lead to also modify other modes -or even make new ones appear- in such a way that the QTM is even larger than before the said modification.

We agree completely with the reviewer. The aim of the single-mode analysis presented in the manuscript is simply to highlight that there may be a positive side to vibronic coupling. We are aware that engineering molecules with desired vibronic properties is an extremely complex task that goes beyond the scope of this work. We have removed the last paragraph from the Discussion section and reformulated significantly our conclusions, in the light of the revised results.

-Regarding the last paragraph "According to the results shown above, the ideal candidates to observe vibronic suppression of QTM (...)":
it happens to be rather poorly elaborated (and so the Conclusions section) in terms of a clear connection to chemical design. Moreover, the proposed requirements which the vibrations should fulfill seem to be even quite demanding. While low-frequency and strongly-coupled-to-the-spin vibrations may help to suppress QTM, they are precisely the ones that contribute to accelerate the thermally-activated relaxation mechanisms.

We accept the criticism of the reviewer. Rather than providing realistic and practical guidelines for chemical design, our intention behind this paragraph centred more around the implications of our model. We have decided to remove it altogether, in an attempt to shift the focus of the manuscript away from chemical design principles.

Reviewer 3:

Authors present a detailed theoretical study on the effects of local vibrations onto the magnetic tunneling in a prototypical case of molecular magnets. The problem is quite subtle and it requires the understanding of many complementary effects (magnetic anisotropy, vibrations, coupling spin phonons, effects of magnetic fields on the quantum tunneling etc.) at the same time. This is done in a very extensive way by the authors who consider the case of $[\text{Dy}(\text{Cpttt})_2]^+$ in details including different configurations of the solvent and for dipolar/hyperfine interactions. That is remarkable. Besides that, authors develop a non-perturbative vibronic approach that allows to treat the problem also in the case of strong spin-phonon coupling regime. This also goes beyond the state of the art. The -indirect- effect of vibrations on the quantum tunneling of magnetization is clearly highlighted and parametrized by the introduction of the measure of the axiality. This aspect is not yet treated in such a detailed way and it represents a clear step ahead in the literature. Method is holistic, it looks correct and sufficiently innovative. Presentation is clear and it duly takes into account previous publications.

We sincerely thank the reviewer for such nice description of the work.

Results are not very surprising and probably not easily exportable to other molecules/ligands.

We respectfully disagree with the reviewer. There has been no detailed discussion of how vibrations affect QTM, especially beyond idealised treatments of the spin-phonon coupling. Our work shows how vibronic effects can be directly assessed. For a more extended discussion, we refer to our reply to the first comment from reviewer 1. Further, we think our work has direct utility to other molecules: we show this in the revised version of this work, where we include results for another Dy(III) complex (see further comments below). Although the results for different SMMs may differ, our method would still be applicable.

The conclusion that molecular engineering could be driven by these results is in my opinion too optimistic and maybe misleading.

We fully accept the reviewer's point, which is in line with the comments from reviewer 2. We have toned down such remarks and focussed our conclusions more on the role of vibrations in the determination of ab initio QTM probabilities.

Overall, my opinion is that this work deserves attention and merits publication.

Few suggestions for the authors:

-Since the complexity of the problem will soon diverge with the size of the molecule, it seems doubtful that this approach can be implanted to other cases, therefore it seems more fair to present this work as "case study" on a prototypical Dy(III) SMM rather than a general approach. Authors should consider this in revising the key sessions of the manuscript (abstract, introduction, conclusions)

Given that first-principles calculations of phonons in molecular systems are now commonplace, our method is directly applicable to such cases. Our polaron model builds on those calculations, adding no further significant computational cost. To support our point, we have expanded the manuscript with an analysis of another prototypical Dy(III) SMM. This shows that our methods can be applied to different SMMs, in amorphous frozen solution environments and in the crystalline phase alike.

- Identify vibrations is the tricky point yet the main suggestion of the work. It would be interesting to spell out the low energy spectrum of the Dy(III) case considered here, for instance by zooming fig.1c (for instance 0-50cm⁻¹). If this zoom looks favorable, it could be interesting to quench "bad" vibrational mode and keep the "good" one by decreasing the temperature and this may stimulate some experiments.

We thank the reviewer for the interesting suggestion. In this revised version, the role of low frequency vibrations is considerably reduced. However, it is still interesting to see if there is any specific correlation between vibrational frequency and impact on vibronic QTM. Below, we show a version of Fig. 4a,c where different frequency ranges are represented with different colours. Only one of the two compounds [Dy(bbpen)Br] shows a clear separation of frequency ranges with regards to the QTM behaviour: vibrations below 100 cm⁻¹ tend to promote QTM, while vibrations between 100 and 200 cm⁻¹ tend to suppress it. However, we stress that the mechanism that we describe in the manuscript does not rely on thermal excitation of the modes and is therefore temperature-independent. The vibronic modulation of the *g*-matrix originates purely from spin-phonon coupling, which generates spin-dependent distortions of the molecular geometry.

Reviewers' Comments:

Reviewer #1:

Remarks to the Author:

Authors addressed all my comments properly and the revised version gained in coherency and clarity. Thus, I suggest to publish this article in its current form.

Reviewer #2:

Remarks to the Author:

The ms. is now improved. I have only some minor comments:

In the footnote of Fig.3, in "(b,c) Distribution of electronic (blue) and vibronic (orange)..." "b" should be changed with "c" and "c" with "d". In the same figure, there are 3 colors: orange, brown and blue, while in the text only orange and blue are mentioned. What the brown color is dealing with?

Reviewer #3:

Remarks to the Author:

Authors replied to all technical criticisms within the assumptions made and, overall, the revised manuscript is now improved. In my opinion, the work represents a valuable effort to consider spin-phonon mechanism in QTM with state-of-the-art computational techniques and, as such, it is an interesting case study, treated ab-initio and in full details -probably- for the first time.

Said that, I agree with other referees that, since the computational problem diverges with the size of the molecule and with the variety of configurations of the external ligands and the environment, the proposed solution is hardly exportable to other cases and the main results are not surprising since there is no chances to control "bad" or "good" phonon modes for QTM, as authors admit. In this respect, I cannot evaluate the impact this work may have in the community and it is probably an Editor's choice to see whether the manuscript deserves publication in Nature Communication.

Reply to Reviewers Comments

Reviewer #1 (Remarks to the Author):

Authors addressed all my comments properly and the revised version gained in coherency and clarity. Thus, I suggest to publish this article in its current form.

We thank the Reviewer for acknowledging the improvements to the manuscript and recommending publication.

Reviewer #2 (Remarks to the Author):

The ms. is now improved. I have only some minor comments:

In the footnote of Fig.3, in "(b,c) Distribution of electronic (blue) and vibronic (orange)..." "b" should be changed with "c" and "c" with "d". In the same figure, there are 3 colors: orange, brown and blue, while in the text only orange and blue are mentioned. What the brown color is dealing with?

We thank the Reviewer for noticing the mislabelling of panels (c,d) in the caption of Fig. 3. The brownish area in the histograms represents the overlap between orange and blue distributions with finite transparency. We have eliminated this possible source of ambiguity by adding bin outlines, as already done in panels a, b.

Reviewer #3 (Remarks to the Author):

Authors replied to all technical criticisms within the assumptions made and, overall, the revised manuscript is now improved. In my opinion, the work represents a valuable effort to consider spin-phonon mechanism in QTM with state-of-the-art computational techniques and, as such, it is an interesting case study, treated ab-initio and in full details -probably- for the first time.

Said that, I agree with other referees that, since the computational problem diverges with the size of the molecule and with the variety of configurations of the external ligands and the environment, the proposed solution is hardly exportable to other cases and the main results are not surprising since there is no chances to control "bad" or "good" phonon modes for QTM, as authors admit. In this respect, I cannot evaluate the impact this work may have in the community and it is probably an Editor's choice to see whether the manuscript deserves publication in Nature Communication.

We thank the Reviewer for the positive remarks on our work. Regarding the Reviewer's comment on applicability to other systems, we have to respectfully disagree. As the Reviewer points out, the complexity of the problem scales with the size of the molecular system, but that is always the case with any quantum chemistry calculation. Our analysis based on a polaron model does not represent a significant additional computational cost on top of the ab initio calculations, and can thus be easily replicated by others. Moreover, we explicitly show that the method can be applied to different molecular systems. We do not see why the same analysis could not be performed on other complexes, provided that the electronic structure, spin-phonon couplings and vibrational modes can be calculated. These types of calculations are now easily accessible and are routinely performed in several groups working on molecular magnetism.

Lastly, we point out that we do not make claims on if/how it would be possible to "control the bad or good vibrations". Our focus here is to discuss and quantify the effect of vibrations on QTM in a rigorous and unambiguous way, using theoretical techniques (polaron transformation) that are not necessarily well known in the molecular magnetism community.